# UNCOVERING ACTIVATION KEYS IN THE DARK: REVEALING LEARNED CONCEPTS IN LORA TEXT-TO-IMAGE MODELS

## ABSTRACT

Low-Rank Adaptation (LoRA) has become a widely adopted technique for customizing large diffusion models, enabling users to inject new styles, characters, or identities into text-to-image generation with minimal computational cost. While this flexibility fuels creative expression, it also opens the door for injecting sensitive or potentially harmful content, such as political figures' faces, copyrighted characters, or explicit imagery, into generative models. These LoRA adapters are often distributed without documentation, making it difficult to identify the concepts they encode or understand how they are triggered. This lack of transparency poses serious challenges for moderation, accountability, and large-scale content auditing in open-source model ecosystems. To address this risk, we adopt the role of a model investigator and introduce the LoRA "activation key" discovery problem: given a suspect LoRA and its base model, identify a text embedding that reliably activates behaviors unique to the LoRA. This activation key serves as a forensic probe to reveal hidden concepts introduced during fine-tuning. To achieve this, we propose a two-stage optimization framework [1]. We first perform an evolutionary search in the token space to identify promising candidate prompts, followed by gradient-based refinement in the embedding space. Our objective encourages the LoRA model to generate concentrated outputs while maximizing divergence from the base model, resulting in an embedding that reveals distinct LoRA-specific behaviors. Experiments on six public LoRA adapters show that the proposed method recovers ground-truth concepts in both white-box and black-box settings. Our work demonstrates the feasibility of LoRA forensics and highlights the need for auditing tools in open-source model ecosystems.

## 1 INTRODUCTION

Text-to-image diffusion models, such as Stable Diffusion, have advanced rapidly, allowing the generation of high-quality images from text prompts as shown by Ho et al. (2020) and further developed by Rombach et al. (2022) and Saharia et al. (2022). In parallel Hu et al. (2022) introduced Low-Rank Adaptation (LoRA) which has emerged as a lightweight and efficient fine-tuning technique that injects novel concepts such as styles, characters, or identities into powerful base models, with minimal training data. This has fueled a wave of community-driven innovation on platforms such as CivitAI by Maier (2022) and Hugging Face by Clément Delangue (2016), where LoRA models are openly shared, customized, and deployed.

However, the same flexibility that enables creative expression also allows users to inject sensitive or harmful content into generative models. For example, LoRA adapters may be fine-tuned to reproduce copyrighted material, impersonate political figures, or generate inappropriate imagery. These models are often shared without any documentation or disclosure of their intended functionality, making it difficult to verify what concepts have been introduced or how they are activated. This lack of transparency poses a serious challenge for content auditing, safety enforcement, and regulatory oversight. Once such models are widely disseminated through open platforms, the resulting impact can be unpredictable and difficult to contain.

---

[1]Code is available at: https://anonymous.4open.science/r/Uncovering-Activation-Keys-in-the-Dark-anon/.

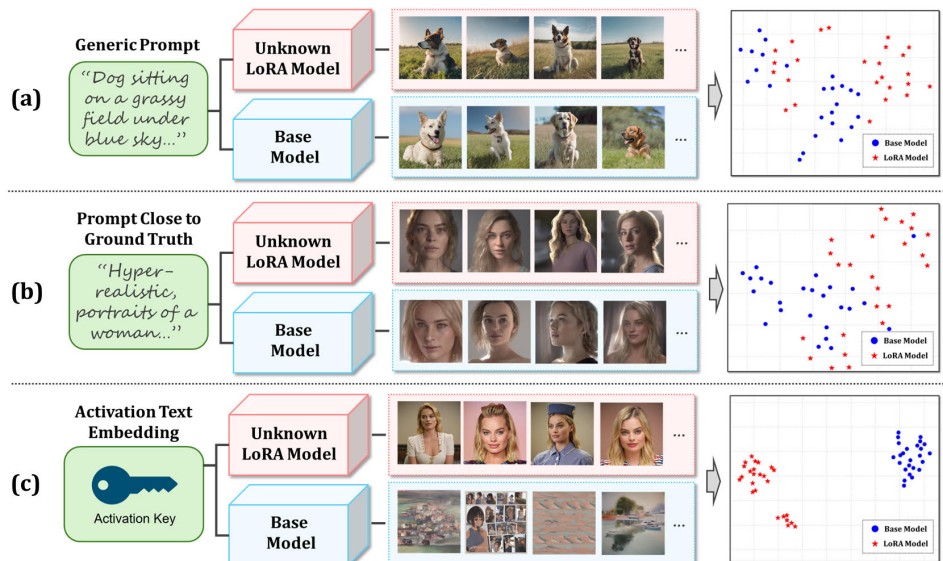

Figure 1: Comparison of prompts for the *Margot Robbie* LoRA, with t-SNE projections of image embeddings of the generated outputs shown on the right. (a) A generic prompt yields nearly identical results, with no separation in the embedding space. (b) A prompt semantically close to the ground truth still fails to reveal LoRA-specific features; generations largely overlap. (c) An optimized activation key reveals the LoRA concept: t-SNE shows a clear separation between LoRA and base output embeddings.

Therefore, enabling the auditing of undocumented LoRA adapters is critical for ensuring account-ability and preventing the unchecked spread of harmful content. In this work, we adopt the role of a forensic investigator, aiming to uncover hidden concepts introduced during fine-tuning. Specifically, given a suspect LoRA and its corresponding base diffusion model, our goal is to determine whether the LoRA encodes previously undeclared concepts, and if so, to identify what those concepts are. This is particularly challenging because such behaviors may not be triggered by typical prompts; instead, they may rely on specific, potentially obfuscated text embeddings to activate. As illustrated in fig. 1, we compare three scenarios using a LoRA trained on the visual identity of *Margot Robbie*, where the hidden trigger word is *"margot_robbie"*. A generic prompt such as *"Dog sitting on a grass field under blue sky"* produces nearly identical outputs from both the base and LoRA models, showing no meaningful separation. Even with a semantically related prompt like *"Hyper-realistic portraits of woman"*, the LoRA still fails to reveal its unique concept. In contrast, when queried with a specific *activation key*, the LoRA consistently exposes the hidden concept while the base model does not, demonstrating that even without knowing the true trigger word, a carefully crafted alternative embedding can still uncover LoRA-specific behavior.

We thus describe our task as the discovery of an activation key, a text embedding that reliably elicits behaviors unique to the LoRA but absent in the base model. Discovering such activation keys is not trivial. Prior approaches, such as the membership inference method proposed by Shokri et al. (2017), assume access to known data samples, which are typically unavailable in real-world LoRA deployments. Other techniques, like weight leakage or model inversion demonstrated by Yao (2024), often require access to model internals or additional training of auxiliary decoders, making them unsuitable for black-box or large-scale auditing scenarios.

To systematically uncover behavioral differences between a suspect LoRA and its base model, we formulate an explicit objective function that guides the discovery of activation keys. This objective is built on two complementary components. The first, intra-model dispersion, encourages the activation key to induce consistent and semantically coherent outputs within the LoRA, while producing diverse or scattered outputs from the base model. The second, inter-model similarity, penalizes the activation key if it results in overlapping or visually similar generations across the two models. The overall goal is to maximize intra-model consistency in the LoRA while minimizing similarity with the base model, thereby exposing distinct behaviors attributable to fine-tuning. Based on such an objective function, we design a two-stage framework to uncover the activation key. The first

stage performs evolutionary search in the token space to identify a pool of candidate prompts as a first approximation. These candidates are then refined in the second stage through gradient-based optimization in the continuous embedding space, allowing fine-grained tuning beyond the limitations of discrete tokens. This combination enables both coarse exploration and precise adjustment, ultimately yielding activation keys that reliably reveal LoRA-specific concepts.

Our main contributions are summarized as follows:

1. We introduce the LoRA activation key discovery problem, which aims to audit undocumented LoRA adapters by uncovering hidden fine-tuned concepts. We highlight the practical importance of this underexplored problem for responsible model sharing and forensic accountability.

2. We propose a two-stage optimization framework that discovers activation keys by maximizing behavioral divergence between the LoRA and its base model. Our approach is model-agnostic and applicable in both white-box and black-box settings.

3. Extensive experiments on publicly available LoRA adapters demonstrate that the proposed method effectively recovers hidden concepts, providing interpretable and reliable evidence of fine-tuned behaviors.

## 2 RELATED WORK

Recent research highlighted that fine-tuned generative models, especially LoRA-adapted diffusion models, pose serious privacy risks. Yao (2024) demonstrated that LoRA weights alone could leak training data, reconstructing sensitive inputs via a variational auto-encoder without requiring prompts or original images. Similarly, Carlini et al. (2023) revealed that diffusion models memorized and regenerated exact training samples, including personal or copyrighted content. These findings suggest that such techniques can be considered as methods to audit LoRA models and uncover hidden concepts. Earlier work on membership inference Shokri et al. (2017) and reconstruction attacks Buzaglo et al. (2023) and Balle et al. (2022) showed related vulnerabilities across architectures. However, these approaches predominantly relied on white-box access or prior knowledge of the training data, and in some cases required extensive training for each LoRA model, limiting their applicability in real-world scenarios.

Beyond privacy risks, research has also shown that prompt optimization can influence how specific concepts are triggered in text-to-image diffusion models, though primarily to improve output quality and coherence. Hao et al. (2023) proposed prompt adaptation, while Mo et al. (2024) introduced prompt auto-editing (PAE), both applying reinforcement learning to adjust prompt wording, weights, and injection timing for better aesthetics and semantic alignment. Human-guided strategies, including genetic algorithms by Pavlichenko & Ustalov (2023) and human-in-the-loop frameworks, refine prompts through iterative feedback. Oppenlaender (2023) identified practical prompt modifiers and characterize prompt engineering as a creative human-computer interaction process, and large-scale evaluations by Liu & Chilton (2022) analyzed prompt composition and keyword effects. Inspired by these techniques, our work adapts the idea of prompt optimization toward uncovering LoRA-specific activation keys.

## 3 SCENARIO

We consider a scenario where an investigator analyzes a suspect LoRA model that is deployed on top of a known base diffusion model but is provided without any accompanying documentation describing its intended purpose. The model takes as input a text prompt (and random seed) and produces as output an image synthesized by the diffusion pipeline. The LoRA may internally encode a hidden concept, a semantic or visual pattern (e.g., a particular object, style, or even inappropriate content) that is not disclosed in the model's documentation but can be elicited through specific triggers. The investigator operates under a white-box assumption, with full access to the parameters of both the base model and the LoRA, the ability to issue arbitrary queries, and the privilege to inspect intermediate signals such as embeddings and logits. Within this setting, the investigator's task is to derive a text embedding, termed an activation key, that when inserted into a prompt consistently triggers the generation of the hidden concept encoded by the LoRA. The activation key serves as the definitive

output of the investigation, providing a concrete mechanism for exposing and validating the LoRA's concealed behavior.

# 4 OBJECTIVE FUNCTION

An activation key refers to a special text embedding whose role is to expose the hidden concept encoded by a LoRA, i.e., the behavioral distinction of the LoRA compared with its base model. To serve this purpose, such an embedding must simultaneously satisfy three requirements: 1) **Consistency within the LoRA.** It should drive the LoRA model to generate outputs that consistently express the same underlying concept. 2) **Diversity within the base model.** Under the same embedding, the base model should generate outputs that vary across concepts rather than concentrating on a single one. 3) **Discrepancy across models.** The outputs of the LoRA and the base model should remain as dissimilar as possible, so that the embedding highlights their difference. Only when these three conditions are met can the embedding be regarded as a valid activation key, capable of constructing a clear distinction between the LoRA and its base model. Based on these principles, we formalize the following objective:

$$f_{\langle \mathcal{L}, \mathcal{B} \rangle}(m) = \{-\alpha \cdot \mathtt{S}_{\mathcal{L}}(m) + \beta \cdot \mathtt{S}_{\mathcal{B}}(m)\} - \{\gamma \cdot \mathtt{IS}_{\langle \mathcal{L}, \mathcal{B} \rangle}(m)\}$$

where $\mathtt{S}_{\mathcal{L}}(m)$ and $\mathtt{S}_{\mathcal{B}}(m)$ denote the intra-model spread for images generated by the LoRA $\mathcal{L}$ and the base model $\mathcal{B}$, and $\mathtt{IS}_{\langle \mathcal{L}, \mathcal{B} \rangle}(m)$ measures the inter-model similarity, with the given embedding $m$. The hyperparameters $\alpha, \beta, \gamma > 0$ control the relative importance of each term, and their individual effects are further examined in the ablation study presented in Section 6.2.2.

## 4.1 INTRA-MODEL SPREAD $\mathtt{S}_{\mathcal{L}}(m), \mathtt{S}_{\mathcal{B}}(m)$

The intra-model spread of a text embedding $m$ with respect to a model $\mathcal{M} \in \{\mathcal{L}, \mathcal{B}\}$ quantifies the variability of the model's outputs under different random seeds. Formally, it is defined as the expected cosine dissimilarity between image embeddings obtained from the CLIP image encoder by Radford et al. (2021) of two images generated from the same embedding $m$ but with independent seeds [2]:

$$\mathtt{S}_{\mathcal{M}}(m) := 1 - \mathbb{E}\left[\cos(\mathtt{CLIP}(\mathcal{M}(m, S_1)), \mathtt{CLIP}(\mathcal{M}(m, S_2)))\right]$$

where $S_1$ and $S_2$ are independently sampled seeds, $\cos(\cdot, \cdot)$ denotes cosine similarity, and $\mathcal{M}(m, s)$ represents the image generated by model $\mathcal{M}$ given text embedding $m$ and seed $s$. Intuitively, a smaller spread $\mathtt{S}_{\mathcal{L}}(m)$ implies that the LoRA consistently generates semantically aligned outputs, which is desirable for revealing its hidden concept. Conversely, a larger spread $\mathtt{S}_{\mathcal{B}}(m)$ indicates that the base model produces more diverse outputs, reinforcing that the embedding does not correspond to a strong concept in the base model.

## 4.2 INTER-MODEL SIMILARITY $\mathtt{IS}_{\langle \mathcal{L}, \mathcal{B} \rangle}(m)$

Likewise, the inter-model similarity is defined as the expected cosine similarity between the CLIP embeddings of images generated by the two models under the same text embedding but with independently sampled seeds:

$$\mathtt{IS}_{\langle \mathcal{L}, \mathcal{B} \rangle}(m) := \mathbb{E}\left[\cos(\mathtt{CLIP}(\mathcal{L}(m, S_1)), \mathtt{CLIP}(\mathcal{B}(m, S_2)))\right]$$

where $S_1$ and $S_2$ are independent random variables representing the seeds. A smaller value of $\mathtt{IS}_{\langle \mathcal{L}, \mathcal{B} \rangle}(m)$ implies greater semantic divergence between the outputs of the LoRA and the base model, thereby isolating concepts that are uniquely attributable to the LoRA.

## 4.3 APPROXIMATION OF $f_{\langle \mathcal{L}, \mathcal{B} \rangle}(m)$

In practice, the expectations in the objective function cannot be computed exactly. Instead, we approximate them using a group of $n$ randomly sampled seeds $S = \{s_1, s_2, ..., s_n\}$. For each model $\mathcal{M} \in \{\mathcal{L}, \mathcal{B}\}$ and seed $s_i$, we denote the corresponding CLIP embedding as

$$e_i^{\mathcal{M}} := \mathtt{CLIP}(\mathcal{M}(m, s_i)).$$

---

[2] *CLIP embeddings* denote image embeddings from the CLIP encoder unless stated otherwise.

Using these embeddings, $f_{\langle \mathcal{L}, \mathcal{B} \rangle}(m)$ are approximated as follows:

$$\mathrm{S}_{\mathcal{M}}(m) \approx \frac{1}{n(n-1)} \sum_{i=1}^{n} \sum_{\substack{j=1 \\ j \neq i}}^{n} \left( 1 - \frac{e_i^{\mathcal{M}} \cdot e_j^{\mathcal{M}}}{\|e_i^{\mathcal{M}}\| \, \|e_j^{\mathcal{M}}\|} \right), \quad \mathcal{M} \in \{\mathcal{L}, \mathcal{B}\}$$

$$\mathrm{IS}_{\langle \mathcal{L}, \mathcal{B} \rangle}(m) \approx \frac{1}{n(n-1)} \sum_{i=1}^{n} \sum_{\substack{j=1 \\ j \neq i}}^{n} \frac{e_i^{\mathcal{L}} \cdot e_j^{\mathcal{B}}}{\|e_i^{\mathcal{L}}\| \, \|e_j^{\mathcal{B}}\|}$$

## 5 TWO-STAGE SEARCH

To identify an activation key that maximizes the objective function defined in Section 4, we design a two-stage search framework (fig.2). The first stage employs an evolutionary search over discrete prompts, where candidate prompts are iteratively refined through mutation and crossover operations to locate a globally promising initialization. The second stage then performs gradient-based optimization in the continuous embedding space, starting from the best prompt obtained in Stage 1 and refining it via gradient ascent. This hybrid strategy combines the global exploration ability of evolutionary search with the fine-grained optimization enabled by differentiable objectives.

### 5.1 STAGE 1: EVOLUTIONARY SEARCH FOR STARTING POINT

In the first stage, we search for a high-quality initialization prompt $p^*$ from a large discrete space of natural language tokens. The vocabulary is drawn from the Brown corpus of NLTK by Bird (2006), and an initial population $P_0$ is formed by randomly concatenating sampled tokens. A fixed set of random seeds $S = \{s_1, s_2, \ldots, s_n\}$ is used across all generations to ensure consistency and reduce stochastic variance. For each generation $g$, all prompts in $P_g$ are evaluated using the objective function $f(\cdot)$ (we use $f(\cdot)$ as shorthand for $f_{\langle \mathcal{L}, \mathcal{B} \rangle}(m)$). Token-level scores are then derived from the performance of the prompts in which they appear, normalized by prompt length:

$$s_t = \frac{f(p)}{\mathrm{len}(p)}, \quad t \in p.$$

Tokens unseen in any prompt are assigned a baseline score of $0.0$. Based on these scores, candidate prompts undergo mutation and crossover:

**Mutation.** Tokens may be added, removed, or replaced. Additions are drawn from high-scoring tokens or their semantic neighbors (using CLIP text embeddings), selected by a softmax distribution over token scores. Low-scoring tokens (below the 20% percentile) are removed, while replacements substitute the lowest-scoring token with a semantic neighbor. **Crossover.** Two parent prompts exchange subsequences: shared tokens are preserved, while unmatched tokens are replaced by semantically similar alternatives. After $G$ generations, the best-performing prompt is selected as

$$p^* = \operatorname*{arg\,max}_{p \in \bigcup_{g=1}^{G} P_g} f(p),$$

which is then used to initialize the second stage.

### 5.2 STAGE 2: OPTIMIZATION OF THE INITIAL ACTIVATION KEY

In the second stage, we refine the initialization $p^*$ by directly optimizing its text embedding. The candidate embedding is initialized as $m^{(0)} = \mathrm{CLIP}(p^*)$, where $\mathrm{CLIP}(\cdot)$ denotes the CLIP text encoder. At each iteration $t$, the objective score $f(m^{(t)})$ is computed using a fixed seed set $S$. Its gradient is obtained via backpropagation and used to update the embedding:

$$m^{(t+1)} = m^{(t)} + \eta \nabla_m f(m^{(t)})$$

where the gradient $\nabla_m f(m^{(t)})$ decomposes into contributions from intra-model spread and inter-model similarity:

$$\nabla_m f(m^{(t)}) = -\alpha \cdot \nabla_m \mathrm{S}_{\mathcal{L}}(m) + \beta \cdot \nabla_m \mathrm{S}_{\mathcal{B}}(m) - \gamma \cdot \nabla_m \mathrm{IS}_{\langle \mathcal{L}, \mathcal{B} \rangle}(m)$$

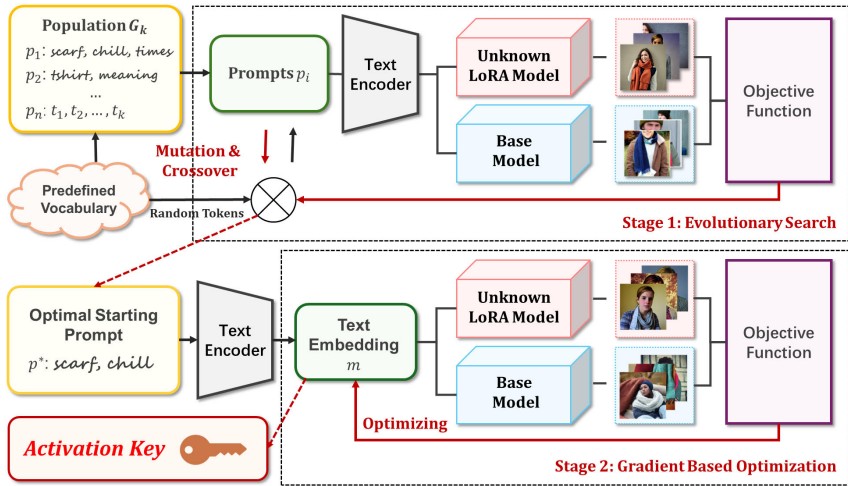

Figure 2: Overview of the proposed two-stage optimization framework for LoRA concept discovery.(1) Evolutionary Search: Prompts are sampled from a vocabulary and refined through genetic operations, with the best candidate chosen by the objective function. (2) Gradient-Based Optimization: The candidate prompt is embedded and optimized via gradient ascent, producing an activation key that reveals the LoRA-specific concept.

The process is repeated until convergence or for a fixed number of steps. The final embedding $m^*$ serves as the activation key, revealing the LoRA-specific concept that does not emerge in the base model.

## 6 EXPERIMENTAL RESULTS

### 6.1 IMPLEMENTATION DETAILS AND EVALUATION METRICS

The proposed method was evaluated using both the Stable Diffusion v1.5 (SD-1.5) by Rombach et al. (2022) checkpoint and Stable Diffusion XL (SDXL) by Podell et al. (2024) as base models, along with open-source, publicly available community-trained LoRA adapters sourced from CivitAI by Maier (2022) used strictly for experimental purposes. The images are generated at a resolution of $512 \times 512$, using 25 diffusion steps for Stable Diffusion v1.5 and 50 steps for SDXL. For all the embedding generations and similarity computations CLIP encoder introduced by Radford et al. (2021) is used. The evolutionary search and gradient refinement procedures are implemented in PyTorch described by Collobert et al. (2011) and executed on a single NVIDIA A100 GPU. Continuous embedding optimization is performed using the Adam optimizer of Kingma & Ba (2017) with a learning rate of $\eta = 10^{-4}$. The final activation key discovered by the proposed method serves as the sole guiding input for generation in both base and LoRA models. Finally, we apply a vision–language model (VLM) introduced by Liu et al. (2024) to LoRA-generated images (conditioned on the discovered activation key) to describe their visual content in natural language, providing an additional layer of semantic evaluation. We evaluate the quality of the recovered activation keys using three complementary metrics. Trigger Embedding Similarity (TrigSim) measures cosine similarity between the recovered and ground-truth trigger embeddings in text space. Caption Similarity (CapSim) compares VLM-generated captions with ground-truth captions to assess semantic alignment. Jayasumana et al. (2024) introduced CLIP-based Maximum Mean Discrepancy (CMMD) which quantifies visual distributional similarity in CLIP space, with lower scores indicating better alignment. The fig. 3 shows the target images for each LoRA model. We also compare against a random prompt baseline to highlight the benefits of structured optimization.

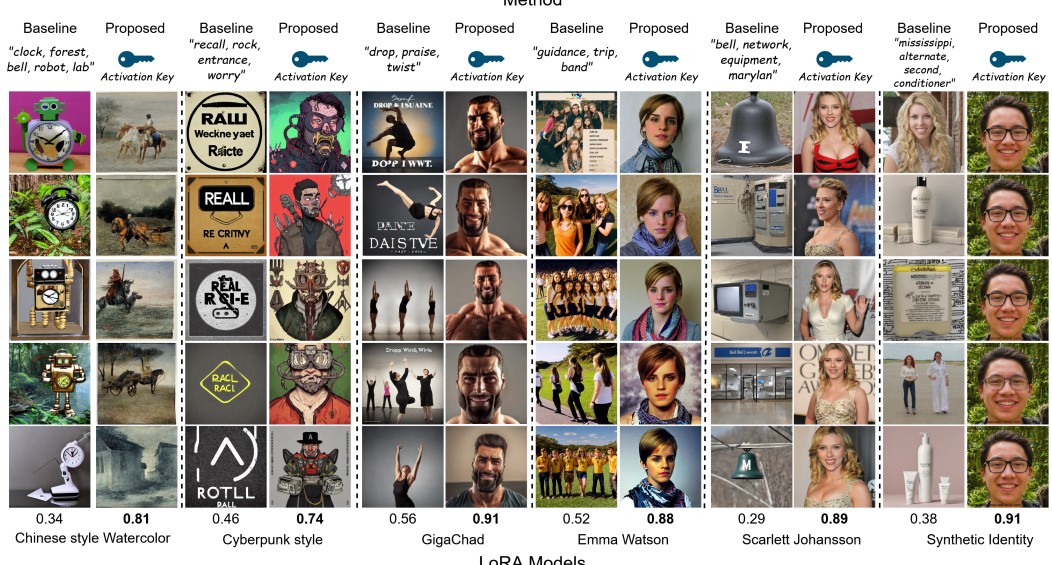

Figure 3: Ground-truth visual references for each LoRA concept used in our experiments. Each group corresponds to a specific concept. The concept "qwerty" is a synthetic identity. The ground truth triggers are mentioned below.

## 6.2 LORA EVALUATION

### 6.2.1 EFFECTIVENESS OF ACTIVATION KEYS FOR FORENSIC ANALYSIS

The proposed method was evaluated on six LoRA models spanning both stylistic and identity-based concepts. As shown in fig. 4, the recovered activation keys consistently reproduce the intended LoRA concepts, whereas the best-performing prompt from the baseline fails to achieve comparable alignment. For style-oriented LoRAs such as Chinese Watercolor and Cyberpunk Anime, our

Figure 4: Comparison of probing strategies across six LoRA models. The top row shows the inputs used for generation: the best-performing baseline prompt and the recovered activation key. Optimization objective scores $f(m)$ are reported for each model, and the closest token approximations of the activation keys are listed in Table 1.

method elicits coherent artistic features that reflect the underlying style, while the baseline outputs remain fragmented and inconsistent. For identity-based LoRAs, the gap is even more pronounced, the baseline generations drift away from the target identity, whereas the recovered activation keys reliably capture distinctive facial traits and semantic cues. Finally, for the Synthetic Identity LoRA, the proposed method maintains stable identity features across outputs, while the baseline continues to wander toward unrelated or noisy content. Overall, these results demonstrate that the proposed structured search is substantially more effective than the baseline method, consistently uncovering hidden LoRA concepts even under challenging identity-focused settings. We further validate these findings in Section 6.3, where quantitative and semantic analyses confirm the superiority of recovered activation keys over baseline prompts.

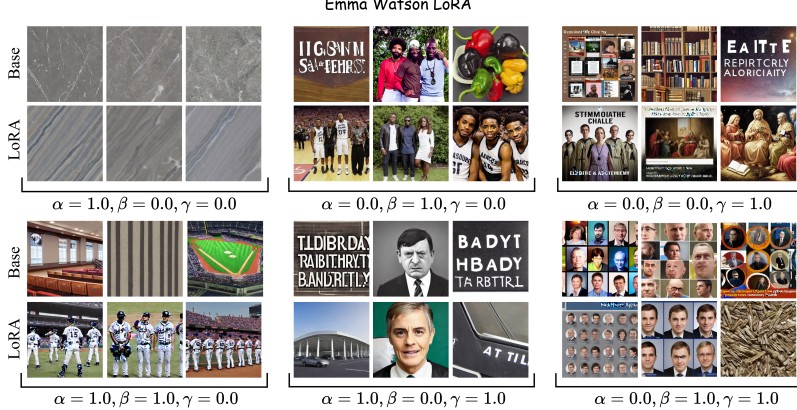

Figure 5: Loss-term ablation across six configurations of $(\alpha, \beta, \gamma)$ excluding the full objective. Each block shows Base (top) and LoRA (bottom) outputs.

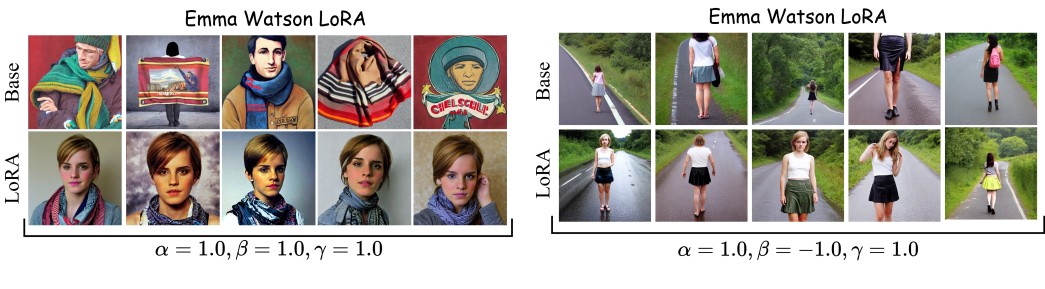

(a) Full objective $(1, 1, 1)$.          (b) Inverted base-spread $(1, -1, 1)$.

Figure 6: Effect of the complete objective vs. inverting the base-spread term. The inverted version collapses base-model outputs toward the LoRA outputs, eliminating the desired asymmetry.

### 6.2.2 ABLATION STUDY ON LOSS TERMS

To understand the role of each component of the objective in Section 4, we ablate the three terms controlled by $\alpha, \beta, \gamma$. We evaluate all seven non-trivial binary configurations:

$$(\alpha, \beta, \gamma) \in \{(1, 0, 0), (0, 1, 0), (0, 0, 1), (1, 1, 0), (1, 0, 1), (0, 1, 1), (1, 1, 1)\},$$

using the same evaluation setup as in Section 6.1. The full objective $(1, 1, 1)$ serves as the reference configuration (fig. 6(a)); the remaining six settings, shown in fig. 5, isolate the effect of enabling individual or paired terms. When only a single term is active $(100), (010), (001)$, optimization becomes overly greedy: it improves the selected component but fails to isolate the LoRA concept. These settings often collapse onto unrelated visual patterns, indicating that none of the three terms alone provides sufficient structure for reliable activation-key recovery. With two terms active $(110), (101), (011)$, optimization improves but remains unstable. The $(1, 0, 1)$ setting illustrates this: although the LoRA concept begins to surface, the signal is weak. Notably, we observe that even though the $S_{\mathcal{B}}$ (base-spread) term is disabled, the base spread still remains relatively high yet suboptimal, since the objective does not explicitly encourage the base model to remain generic. Empirically, results almost always improve when the $S_{\mathcal{B}}$ is included. To further understand the importance of the $S_{\mathcal{B}}$, we also test an inverted variant $(\alpha, \beta, \gamma) = (1, -1, 1)$, which encourages both LoRA and base spreads to be small. As shown in fig. 6(b), this collapses the base and LoRA model outputs toward a nearly identical concept, breaking the asymmetric behavior the objective is designed to enforce. Overall, these ablations show that all three terms are necessary: minimizing LoRA spread $S_{\mathcal{L}}$, maximizing base spread $S_{\mathcal{B}}$, and minimizing inter-model similarity $\text{IS}_{\langle \mathcal{L}, \mathcal{B} \rangle}$. Removing or inverting any one of them weakens the asymmetric behavior required to reliably expose LoRA-specific concepts, consistent with the observations and intuition used to define the objective in Section 4.

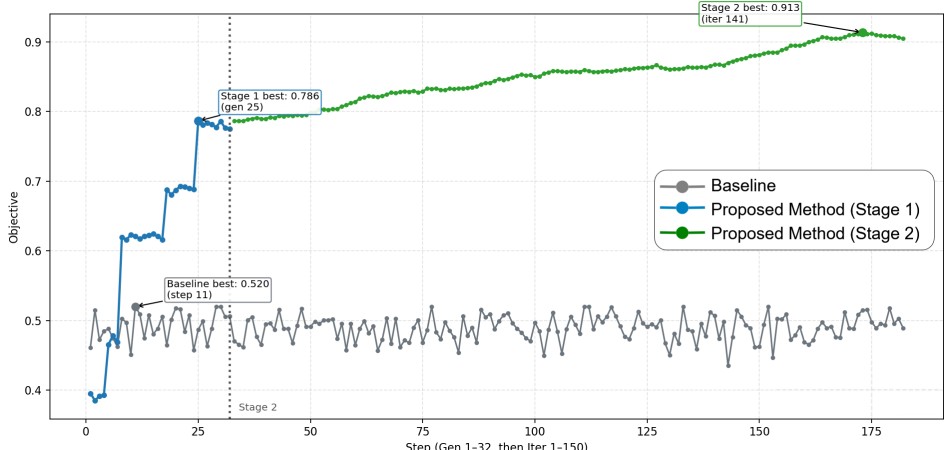

Figure 7: Comparison of optimization trajectories. The baseline fluctuates with no improvement, while Stage 1 and Stage 2 progressively increase the objective.

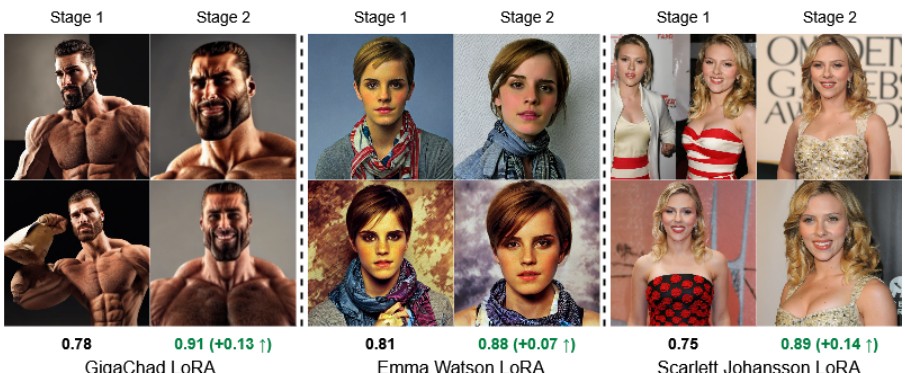

Figure 8: Comparison of Stage 1 and Stage 2 across three LoRAs. Stage 2 consistently improves $f(m)$ and alignment with the target concept.

### 6.2.3 IMPACT OF STAGE 1 AND STAGE 2 OPTIMIZATION

Fig. 7 compares the optimization dynamics of our method against the baseline. The baseline fluctuates without meaningful gains, while Stage 1 rapidly improves, typically reaching scores of 0.7–0.8 before plateauing. Since Stage 1 only requires black-box access, where the investigator can query the text-to-image model with prompts and observe outputs, this shows that reasonable activation keys can be found without internal model access, although with limited performance. Stage 2, which leverages white-box access, consistently builds upon Stage 1 by further refining outputs and aligning them more closely with the ground-truth concept. As shown in Fig. 8, Stage 2 steadily increases the optimization score and produces more coherent and semantically accurate generations across LoRA models. Representative failure cases, where Stage 1 stagnates in low-scoring regions and Stage 2 cannot fully recover are analyzed in Appendix A.2. We also conduct a detailed ablation comparing Stage 1 only, Stage 2 only, and the full two-stage pipeline; these results are reported in Appendix A.3.

### 6.2.4 OPTIMIZATION TRAJECTORIES ACROSS ITERATIONS

The fig. 9 visualizes how activation keys evolve during optimization using t-SNE projections. The baseline remains scattered across all steps with no meaningful structure. In contrast, proposed method shows clear progression where the first stage initiates cluster separation between base and LoRA outputs, while the second sharpens this divide, ultimately converging around the LoRA-specific concept. This trajectory highlights how the two-stage search incrementally aligns the activation key with the target concept.

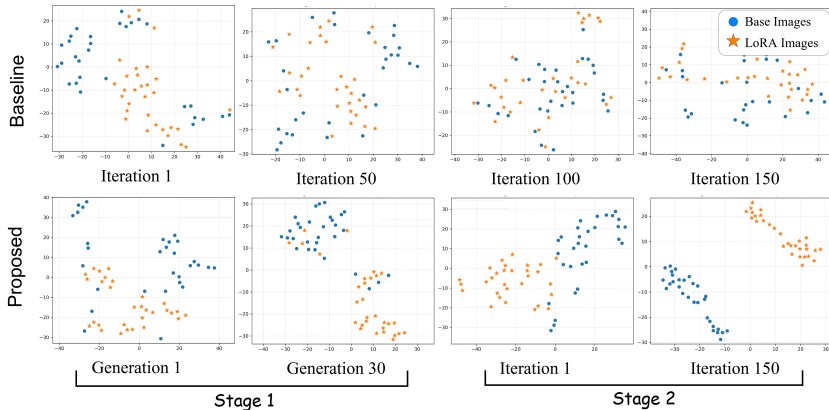

Figure 9: t-SNE evolution across optimization. The baseline (top) remains scattered with no separation, while the proposed method (bottom) progressively clusters LoRA (orange) and base (blue) samples, with Stage 2 yielding a clear LoRA-specific cluster.

Table 1: Baseline prompts vs. recovered activation keys evaluated against ground-truth (GT) triggers. Higher TrigSim and CapSim, and lower CMMD, indicate better recovery. Full VLM captions are provided in Appendix A.5.

| LoRA Model GT Trigger | Method | Prompt / Key (approx.) | VLM Caption (short) | Semantic Consistency | | Qualitative Score |
|---|---|---|---|---|---|---|
| | | | | TrigSim ↑ | CapSim ↑ | CMMD ↓ |
| **Chinese Watercolor** *shuicai_v1* | Baseline | *clock, forest, bell, robot, lab* | Robots, clocks | 0.24 | 0.43 | 0.38 ± 0.017 |
| | Proposed | *listening, let, warmth, come, cavalry* | Chinese watercolor rider | **0.67** | **0.79** | **0.14** ± 0.02 |
| **Cyberpunk Anime** *cyberpunk_anime* | Baseline | *recall, rock, entrance, worry* | Logos, typography | **0.32** | 0.34 | 0.49 ± 0.022 |
| | Proposed | *victor, bastard, academic, farming* | Cyberpunk portraits | 0.31 | **0.76** | **0.12** ± 0.017 |
| **GigaChad** *gigachad* | Baseline | *drop, praise, twist* | Fitness posters | 0.39 | 0.34 | 0.62 ± 0.021 |
| | Proposed | *man, face, champion* | Hyper-muscular male | **0.51** | **0.82** | **0.06** ± 0.012 |
| **Emma Watson** *watson* | Baseline | *guidance, trip, band* | Group photos | **0.42** | 0.52 | 0.38 ± 0.017 |
| | Proposed | *scarf, chill, nelson, times, courage* | Emma Watson portraits | 0.37 | **0.79** | **0.09** ± 0.013 |
| **Scarlett Johansson** *scarlett_johansson* | Baseline | *bell, network, equipment, marylan* | Abstract artworks | 0.26 | 0.28 | 0.68 ± 0.02 |
| | Proposed | *ohwx, enforcement, silly* | Scarlett Johansson portraits | **0.52** | **0.83** | **0.05** ± 0.01 |
| **Synthetic Identity** *qwerty* | Baseline | *mississippi, alternate, conditioner* | Skincare, lifestyle | 0.18 | 0.25 | 0.57 ± 0.019 |
| | Proposed | *alex, progress* | Smiling man with glasses | **0.61** | **0.77** | **0.11** ± 0.015 |

## 6.3 QUANTITATIVE AND SEMANTIC EVALUATION

Table 1 presents the evaluation results for all LoRA models using the metrics from Section 6.1. It includes VLM-generated captions and quantitative scores. The "Prompt / Key" column shows the best Stage 1 prompt, used as a discrete approximation for the proposed method. However, for all experiments, the actual activation key is used which is the continuous text embedding refined in Stage 2. In all cases, recovered activation keys consistently outperform baseline prompts, validating their effectiveness in uncovering LoRA-specific concepts.

## 7 CONCLUSION AND FUTURE WORK

This paper addresses the problem of auditing undocumented LoRA adapters by focusing on the discovery of hidden concepts. We formally define the activation key discovery problem, where the goal is to identify a text embedding that exposes the behavioral distinction of a LoRA compared to its base model. To capture this notion, we introduce an objective function that integrates intra-model spread and inter-model similarity. Building on this formulation, we propose a two-stage search framework, combining evolutionary search for global exploration with gradient-based refinement in the embedding space. Experimental results demonstrate that our framework effectively uncovers LoRA-specific concepts that remain absent in the base model, providing a principled approach to forensic analysis of generative adapters.

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

# A APPENDIX

## A.1 LLM USAGE STATEMENT

A large language model (LLM) was used as an assistive tool for the writing and editing of this paper. Its primary function was to polish and rephrase sentences for improved clarity, readability, and academic style. The authors have reviewed and take full responsibility for all content presented.

## A.2 ANALYSIS OF FAILURE CASES AND LIMITATIONS IN TWO-STAGE OPTIMIZATION

The fig. 10 illustrates representative failure patterns. These arise when poor initial seeds in Stage 1 lead to low-scoring regions that the evolutionary process cannot escape, which in turn limits the effectiveness of Stage 2. In some cases, Stage 1 stagnates at suboptimal scores, while in others Stage 2 adds little or no improvement. In rarer cases, both stages plateau with minimal progress. These results highlight the sensitivity of the pipeline to initialization and the dependence of Stage 2 on Stage 1's search quality. While less frequent than successful runs, they emphasize the need for improved exploration strategies to mitigate cascading failures across the two stages.

The root cause of these failures usually lies in the dependence on initial random seeds in Stage 1. Poor seeds lead to incoherent token combinations or low-quality embedding regions, which Stage 2 cannot meaningfully refine. Although this stochasticity is unavoidable, practical mitigations include increasing the number of images since a larger pool of initial seeds increases the likelihood of finding a promising candidate, reducing the number of steps per run to quickly test candidate activations, and adjusting the exploration parameters of Stage 1 to encourage broader discovery. Importantly, even when individual runs fail, rerunning with different seeds generally produces viable activation keys. Quantitatively, failure cases were not dominant but occurred with some regularity. Across experiments, approximately 14% of runs stalled during Stage 1 and failed to escape low-scoring regions, while an additional 8% exhibited weak improvement in Stage 2 despite a reasonable Stage 1 output.

Finally, regarding computational cost, all experiments were conducted with sequential image generation, though the process is trivially parallelizable. Runtime depends primarily on the number of diffusion steps and the number of images sampled per prompt. On average, each image took approximately one minute to generate on an NVIDIA A100 GPU. With $n$ prompts and $G$ generations, Stage 1 requires generating up to $10 \times n \times G$ images (five from the base model and five from the LoRA per prompt). However, images were only generated when a prompt underwent mutation; otherwise, we reused a memoized list of previously computed scores to avoid redundant evaluations. Stage 2 then refines over 150 iterations, each step involving 10 images. While this makes the

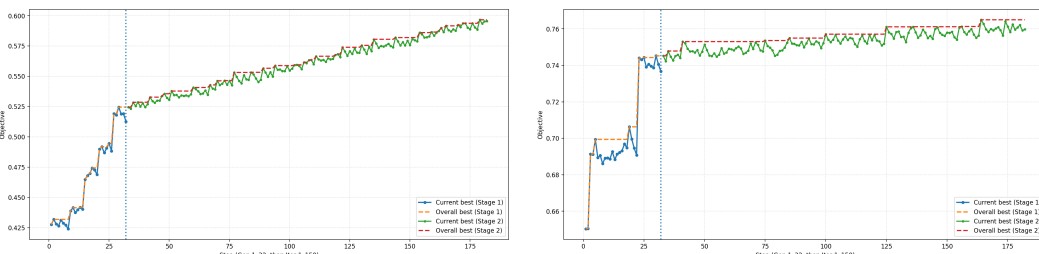

(a) Stage 1 stalls; Stage 2 modest. Stage 1 rises only from ∼0.39 to ∼0.50; Stage 2 reaches only ∼0.60.

(b) Stage 2 fails to improve. Stage 1 reaches ∼0.74; Stage 2 adds at most ≤0.02, saturating near ∼0.76.

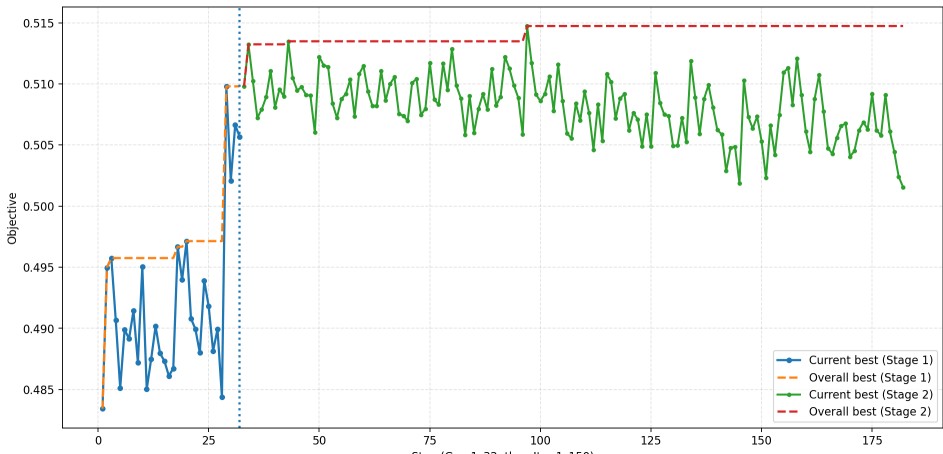

(c) Both stages fail. Stage 1 plateaus near ∼0.50; Stage 2 hovers around ∼0.51 without meaningful gains.

Figure 10: Failure modes of two-stage optimization. (a) Stage 1 stagnates and Stage 2 yields only minor gains. (b) Stage 2 fails to improve on a reasonable Stage 1 result. (c) Both stages fail to progress. Plots show current-best and overall-best trajectories, with Stage 1/2 boundary at step 32.

current implementation computationally demanding, runtime can be substantially reduced through parallelization, memoization, and more efficient diffusion schedules.

### A.3 ABLATION OF STAGE 1 AND STAGE 2

To better understand how the two optimization stages contribute to activation-key discovery, we evaluate four configurations: (i) the random baseline, (ii) Stage 1 only (evolutionary search), (iii) Stage 2 only (gradient refinement from a random initial embedding), and (iv) the full two-stage pipeline. As shown in Table 2, Stage 1 alone already uncovers useful prompts in a purely black-box setting, while Stage 2 alone remains sensitive to initialization. The full two-stage method consistently achieves the highest objective values and successfully reveals LoRA-specific behavior for all models.

### A.4 CHOICE OF LORAS FOR EXPERIMENTS AND MODEL-AGNOSTIC DESIGN

We selected LoRA adapters that reflect both identity-focused and stylistic concepts to evaluate the generality of our approach. Specifically, we chose three identity-based LoRAs that were publicly available, alongside two style-oriented LoRAs. This mixture ensured that our framework was not biased toward a single type of concept, and indeed we observed consistent recovery across both categories. In addition, we conducted further tests on other community LoRAs and LoRAs fine-tuned on private identity datasets, which showed similar trends and confirmed that the method generalizes beyond the main experimental set.

Table 2: Ablation of Stage 1 and Stage 2. We report the average maximum objective value $f(\cdot)$ and the number of LoRAs (out of six) for which a distinct LoRA-specific concept is successfully revealed.

| Configuration | Avg. Max $f(\cdot)$ | Success |
|---|---|---|
| Random Baseline | $0.52 \pm 0.05$ | 0/6 |
| Stage 1 Only | $0.78 \pm 0.04$ | 4/6 |
| Stage 2 Only | $0.61 \pm 0.08$ | 2/6 |
| Stage 1 + Stage 2 (Full) | $\mathbf{0.91 \pm 0.04}$ | **6/6** |

Table 3: Full VLM-generated captions for LoRA-generated images conditioned on recovered activation keys.

| LoRA Model GT Trigger | Method | Prompt / Key (approx.) | VLM Caption |
|---|---|---|---|
| **Chinese Watercolor** *shuicai_v1* | Baseline | *clock, forest, bell, robot, lab* | Toy-like robots, analog clocks, and mechanical instruments in staged real-world settings appearing repeatedly. |
| | Proposed | *listening, let, warmth, come, cavalry* | Traditional Chinese-style watercolor scenes of riders, carriages, and rural life with muted tones. |
| **Cyberpunk Anime** *cyberpunk_anime* | Baseline | *recall, rock, entrance, worry* | Minimalist logos and distorted text elements with no characters. |
| | Proposed | *victor, bastard, academic, farming* | Anime-style portraits in a cyberpunk setting with visible tech augmentations, wires, masks, and dystopian gear. |
| **GigaChad** *gigachad* | Baseline | *drop, praise, twist* | Fitness-themed posters and dance poses with varied text and motion. |
| | Proposed | *man, face, champion* | A hyper-muscular male figure resembling the "GigaChad" meme. |
| **Emma Watson** *watson* | Baseline | *guidance, trip, band* | Group photographs of young women outdoors in casual settings. |
| | Proposed | *scarf, chill, nelson, times, courage* | Portraits of a woman resembling Emma Watson wearing patterned scarves and neutral expressions. |
| **Scarlett Johansson** *scarlett_johansson* | Baseline | *bell, network, equipment, marylan* | Fantasy-style digital artworks and abstract colorful patterns. |
| | Proposed | *ohwx, enforcement, silly* | Scarlett Johansson in various event photos — red carpet and media appearances with consistent hairstyle and makeup. |
| **Synthetic Identity** *qwerty* | Baseline | *mississippi, alternate, second, conditioner* | Mixed visuals including a blonde woman, cosmetic products, and skincare packaging in studio and outdoor scenes. |
| | Proposed | *alex, progress* | Portraits of a smiling young Asian man with glasses, in natural outdoor lighting and consistent framing. |

Importantly, the proposed framework is model-agnostic, particularly in Stage 1. Since the evolutionary search requires only black-box access to generate and score outputs, it can be applied to any text-to-image model without requiring weight inspection. Stage 2, while leveraging gradient-based refinement for improved performance, can also be adapted to different model implementations, with only minor modifications needed to expose gradients for optimization. This separation of stages underscores both the flexibility and portability of the framework across diverse architectures.

### A.5 VLM-GENERATED CAPTIONS FOR CONCEPT ANALYSIS

To assess the effectiveness of recovered activation keys, we used a vision-language model (VLM) to describe LoRA-generated image grids. The VLM was asked: *"What is the main concept represented across these images?"* Table 3 shows the raw captions, which were used to evaluate semantic similarity and help validate whether the recovered keys reflect the intended concept.

These captions provide an additional layer of qualitative evidence complementary to our quantitative metrics. For baseline prompts, the VLM frequently described outputs in vague or inconsistent terms (e.g., "logos," "abstract artworks," or "cosmetic products"), indicating that the LoRA-specific concept was not reliably activated. In contrast, the captions for images generated with our recovered activation keys consistently referenced the intended style or identity. This demonstrates that the proposed method not only aligns with ground-truth concepts in embedding space but also produces semantically coherent outputs recognizable to an external model.

