# OpenReview forum: "Uncovering Activation Keys in the Dark: Revealing Learned Concepts in LoRA Text-To-Image Models"
_ICLR.cc/2026/Conference — Submitted to ICLR 2026_

### Official Review · Reviewer_ffWZ · 2025-10-16

**Soundness:** 4
**Presentation:** 3
**Contribution:** 3
**Rating:** 6
**Confidence:** 4

**Summary:**

This paper tackles the problem of uncovering hidden concepts in LoRA-fine-tuned diffusion models. It introduces a contrastive objective with three loss terms—intra-LoRA consistency, intra-base diversity, and inter-model dissimilarity—and a two-stage optimization framework to recover activation keys. Stage 1 conducts a black-box evolutionary search over discrete prompts, while Stage 2 refines the discovered embedding through gradient ascent. Experiments on several public LoRA adapters with Stable Diffusion 1.5 / SDXL show that the method performs effectively within the tested settings.

**Strengths:**

I genuinely like the paper’s methodological design — it is well-motivated, logically structured, and technically sound.

1. The paper is clearly written and easy to follow, with well-organized sections and informative visualizations.

2. The three-term objective (intra-LoRA consistency, intra-base diversity, and inter-model dissimilarity) is intuitive, elegant, and well grounded in the goal of contrasting LoRA and base behaviors.

3. The two-stage design—evolutionary search followed by gradient-based refinement—is a smart and practical solution to the difficulty of optimizing discrete prompts.

**Weaknesses:**

While the paper is well written and conceptually solid, the evaluation part feels somewhat limited.

1. The experimental scale is relatively small—only six LoRA adapters and two base models (SD 1.5 and SDXL). Expanding to more diverse concepts or datasets, such as the DreamBooth dataset (might need to incorporate some automatic captioning process) or larger community LoRA collections, would better demonstrate robustness and generality.

2. The current comparison baseline (a random or heuristic prompt) is weak. Incorporating stronger baselines such as prompt inversion or optimization methods  would make the empirical claims more convincing.

3. The three loss components are central to the method, yet no ablation study or sensitivity analysis is provided. Evaluating the effect of removing or reweighting each term would clarify their relative importance.

4. It would be valuable to test the approach on newer diffusion architectures (e.g., SD 3 or Flux) to assess whether the mechanism generalizes.

Overall, the proposed method is insightful, and if its with stronger empirical validation, I would lean toward acceptance.

**Questions:**

Apart from the weaknesses mentioned above, I also wonder:


Could the proposed framework be extended to more general fine-tuning settings, such as full-parameter DreamBooth?

---

> ### Author Response · Authors · 2025-11-20
>
> We thank the reviewer for the thoughtful and encouraging assessment of our work. We appreciate the positive evaluation of the methodological design, the clarity of presentation, and the motivation behind our three-term contrastive objective and two-stage optimization framework. Below we respond to each of the concerns raised.
>
> **On experimental scale.**
> We agree that a larger evaluation would further strengthen empirical claims. However, because this is a newly introduced problem, our primary goal in this work is to first investigate whether activation–key forensic analysis is feasible at all under realistic conditions. Since it is not known whether such forensic recovery can be performed, the purpose of our evaluation is to demonstrate feasibility and generalizability across qualitatively different LoRA behaviors rather than to exhaustively enumerate large numbers of similar adapters. Our current set of six LoRAs was therefore selected deliberately to span the principal categories encountered in forensic auditing such as identity-based, stylistic, and character-based LoRAs, so as to test whether the proposed method generalizes across structurally distinct types of concepts. We also reproduced our experiments on several additional in-house identity LoRAs (not publicly distributable), and the method exhibited the same recovery patterns reported in Section~6. Because this problem is new, no large-scale curated auditing benchmark currently exists. Expanding such datasets is an important direction for future work and will allow the community to further validate and compare forensic methods as richer benchmarks are developed.
>
> **On baseline strength.**
> We appreciate the reviewer’s request for stronger baselines. However, existing approaches such as prompt inversion, optimization-based embedding recovery, membership inference, or weight-leakage methods rely on assumptions that are incompatible with the activation-key discovery setting. These techniques typically require access to target identities, training samples, or explicit inversion models none of which are available in our forensic scenario, where both the concept and the trigger are unknown. Under our threat model, these methods effectively collapse to human-style evaluation, which is equivalent to uninformed prompt search. Our random search baseline at each iteration samples an entirely new random prompt and repeats this process for the same number of iterations as our method, ensuring a fair and comparable evaluation. For these reasons, the random search baseline is the most realistic and appropriate comparator for this problem.
>
> **On ablation of the three loss components.**
> We thank the reviewer for highlighting this point. We have added a full ablation study of the three objective terms in the updated version of the paper (Section 6.2.2). We previously omitted it because Section 4 provides the conceptual grounding for the three-term formulation and we thought that was good to give the intuition behind the loss terms, but we agree that explicit empirical ablations substantively strengthen the contribution.
>
> **On generalization to newer diffusion architectures.**
> We appreciate this thoughtful suggestion. The proposed framework interacts only with model outputs and CLIP embeddings and does not rely on architectural properties specific to SD 1.5 or SDXL. Our method uses the behavioral asymmetry between the LoRA model and its base which applies equally to any finetuned–base model pair. For this reason, we expect the method to transfer naturally to modern architectures such as SD~3 or Flux.
>
> **On extension to full-parameter DreamBooth.**
> We thank the reviewer for raising this question. The framework extends naturally to full-parameter finetuning settings (e.g., DreamBooth) because the method does not rely on any structural assumptions unique to LoRA adapters. The procedure contrasts the generative behavior of a finetuned model against its base counterpart, a principle that applies equally to LoRA, DreamBooth, and other PEFT or full-parameter finetuning methods. DreamBooth models may encode more entangled or multi-faceted concepts, potentially requiring multiple activation keys or additional regularization, but the underlying mechanism remains applicable. Adapting the framework to more expressive finetuning regimes is a promising direction for future investigation.
>
> We sincerely thank the reviewer once again for the positive evaluation and constructive feedback. We hope our responses address all concerns, and we would be very happy to provide any additional clarification if needed.

---

### Official Review · Reviewer_BCTv · 2025-10-26

**Soundness:** 2
**Presentation:** 3
**Contribution:** 3
**Rating:** 4
**Confidence:** 5

**Summary:**

This paper introduces the **LoRA activation key discovery problem**, a novel forensic approach for identifying hidden or undocumented concepts within T2I LoRA. Motivated by concerns over moderation and accountability in open-source generative ecosystems, the authors propose a **two-stage optimization framework** consisting of (1) **evolutionary search** in token space and (2) **gradient-based refinement** in embedding space. The objective maximizes behavioral divergence between the LoRA and base models using CLIP-based inter/intra-model similarity measures. Experiments on six public LoRA adapters show that the method effectively recovers ground-truth triggers and reveals distinct LoRA-specific behaviors, verified quantitatively (TrigSim, CapSim, CMMD) and semantically via VLM analysis.

**Strengths:**

- **Novelty:** Defines the previously unexplored area of activation key discovery in LoRA models.

- **Technical Depth:** Well-founded objective and two-stage optimization pipeline, combining discrete and continuous strategies.

- **Empirical Breadth:** Evaluation on multiple LoRAs with robust metrics and semantic validation.

- **Relevance:** Addresses growing safety, transparency, and forensic auditing challenges in T2I ecosystem.

- **Presentation:** Strong writing, clear figures, and reproducible experiments.

**Weaknesses:**

- **Scalability:** The procedure’s computational cost and time may hinder large-scale deployment.

- **Limited Soundness:** The author present that "Our approach is model-agnostic and applicable in both white-box and black-box settings" while actually the proposed method is a two-stage method that relies on the initialization of a black-box token-level optimization for a white-box optimization, which is a white-box method. Results in Fig.5 and Fig.6 are not enough to demonstrate the proposed method is effective in white-box and black-box scenarios (Fig.6 should be organized as only stage1, only stage2 with random initialzed embedding and stage1+stage2 to demonstrate the improvement).

- **Confusing Objectives:** The objective function of "Consistency within the LoRA", "Diversity within the base model" and "Discrepancy across models" is confusing without any rationale. First, some LoRAs are highly semantic-align with their corresponding trigger words which means that the LoRAs can actually replace by a prompt suffix, disaligning with the objective "Diversity within the base model". Second, why should there be "Discrepancy across models" if the LoRAs themselves do not introduce large semantic change to the base LoRA. There are all kinds of LoRAs, some of which might only be used for fine-grained adjustments. Third, "Consistency within the LoRA" is also not grounded, since for style-based LoRAs, CLIP might not capable to extract these features. I recommand the author clarify the audit scope (what types of LoRA) of their method.

- **Ablation Study:** There are no ablation studies for the proposed three objectives and I wonder if they really contribute to the optimization for the above mentioned scenarios.

- **Lack of Multi-task Discovering:** An adaptive attacker can easy design a multi-task LoRAs evasion attack by hidden a malicous task with complex trigger words into a benign task with simple trigger words. The benign task acts as a trapdoor to hijack you optimization.

**Questions:**

Refer to **weakness**. The proposed scenario for LoRA auditing is promising and if the author can address some of my concerns above, I am willing to raise my rating anytime.

---

> ### Author Response · Authors · 2025-11-20
>
> We sincerely thank the reviewer for the detailed and constructive assessment of our submission. We appreciate the recognition of the novelty of the activation–key discovery problem, the technical depth of our two-stage framework, and the relevance of this work to LoRA forensics. We address each concern below.
>
> **On computational cost.**
> We note that the main runtime cost comes from evaluating multiple seeds per embedding, which is naturally parallelizable and can be further reduced with optimizations such as distributed search, faster diffusion solvers, or caching. In forensic auditing, accuracy matters more than speed, and investigators typically analyze only a few suspect LoRAs. Thus, the current runtime is practical, and the remaining overhead reflects implementation details rather than a fundamental limitation of the method.
>
> **On Limited Soundness.**
> As noted in Sec. 3, our framework is designed for full forensic settings where white-box access is available, but the two stages are intentionally separated to support both access modes. Stage 1 is entirely black-box, requiring only text-to-image queries and no access to weights, gradients, or internal signals. Stage 2 is a white-box refinement step used only when gradient access is available. As shown in Sec.6.2.3 and Fig.8, Stage 1 alone often produces meaningful LoRA-base separation, while Stage 2 further strengthens alignment and objective performance. To make this distinction explicit, we added a Stage 1 and. Stage 2 ablation in Appendix A.3. These results show that Stage 1 provides a usable black-box auditing signal, whereas Stage 2 delivers the strongest and most consistent results when white-box access is present. Many practical deployments (e.g., CivitAI or HuggingFace Spaces) do provide checkpoints, making the full method applicable.
>
> **On the rationale and audit scope of the objectives.**
> We thank the reviewer for highlighting the need to clarify the motivation behind the three components of $f_{\langle L,B\rangle}(\cdot)$ (Sec.4). The goal of activation-key discovery is to isolate behavior introduced by the LoRA from that already present in the base model. The three terms each enforce a key aspect of this separation: (1) Low LoRA spread ($S_{\mathcal{L}}$) ensures that a valid activation key consistently evokes the same LoRA-specific concept across seeds; without it, the search collapses to noisy or drifting embeddings. (2) High Base spread ($S_{\mathcal{B}}$) ensures the embedding does not activate a stable concept in the base model, which is essential for defining a LoRA-specific key. (3) Low inter-model similarity ($IS_{\langle L,B\rangle}$) amplifies the behavioral differences introduced by the LoRA, including subtle identity shifts that generic suffix prompts cannot capture. Although some style LoRAs share partial semantics with the base model, our experiments show that this three-term objective reliably separates both identity and stylistic LoRAs.
>
> **On ablation of the three objectives.**
> We thank the reviewer for highlighting this point. We have added a full ablation of loss terms in Section 6.2.2 of the revised version. Although Sec.4 already explains the intuition behind the three terms, we agree that explicit empirical results strengthen the contribution.
>
> **On multi-task LoRAs and adaptive evasion.**
> In this work, our primary goal is to establish that activation-key forensic analysis is feasible under realistic conditions, given that the problem is new and challenging due to the lack of knowledge about the trigger, training data, or underlying concept. Adversarial or multi-task LoRAs designed to evade auditing are a natural direction for future work. We note, however, that our method already handles obfuscated and non-semantic triggers. Beyond the “qwerty” case (LoRA 6), we also trained a LoRA with a fully random trigger, **rNbFUa06LF**, using a synthetic identity from *“This Person Does Not Exist.”* The ground-truth identity image is available **[here](https://this-person-does-not-exist.com/en/download-page?image=gen38f3f5d02f18841ac9353f8db9445797)**. The recovered activation key reliably elicited this hidden identity, as shown in **[Random LoRA Output](https://postimg.cc/R3Nbd80y)**. We also expect that creating evasive multi-task LoRAs would often degrade generative quality or consistency, increasing the attacker’s cost. As in many defensive settings, raising the difficulty of successful evasion is itself valuable. If attackers eventually design LoRAs specifically to evade our method, that would indicate the proposed methods effectiveness against current LoRA designs. Exploring such adversarial scenarios is an interesting direction for future work.
>
> We thank the reviewer once again for the thoughtful feedback. We believe that our clarifications and additional experiments address the concerns, and we are happy to provide any further details.

---

> > ### Comment · Reviewer_BCTv · 2025-11-28
> >
> > Thanks for the author's response.
> >
> > > Investigators typically analyze only a few suspect LoRAs
> >
> > From platform's perspective, I'd like to clarify that adversary could also hide malicious functionality [1] while only offer the benign trigger words and preview images on Civitai. Therefore, every LoRA needs an auditing. Computational cost and time realy matter.
> >
> >
> > > On the rationale and audit scope of the objectives
> >
> > The author provides the **goals** of each terms in the loss function while neglect my questions: (1) some LoRAs are highly semantic-align with their corresponding trigger words which means that the LoRAs can actually replace by a prompt suffix; (2) what if the LoRAs themselves do not introduce large semantic change to the base LoRA, since some of LoRAs are only used for fine-grained adjustments; (3) since for style-based LoRAs, CLIP might not capable to extract these features
> >
> > > On multi-task LoRAs and adaptive evasion
> >
> > I'd like to clarify that LoRA merging is a common practice and merging two LoRAs with different tasks would't change benign LoRAs' functionality significantly [1].
> >
> >
> > - [1] Dong, T., Xue, M., Chen, G., Holland, R., Meng, Y., Li, S., ... & Zhu, H. (2023). The philosopher's stone: Trojaning plugins of large language models. arXiv preprint arXiv:2312.00374.

---

> > > ### Author Response · Authors · 2025-12-03
> > >
> > > We sincerely thank the reviewer for the additional comments and we are happy to provide clarifications.
> > >
> > > **Computation cost and practical feasibility.**
> > > We agree that our method can also be used in the incremental or multi-round auditing manner described by the reviewer. As noted earlier, the computation cost can be reduced substantially through batching, parallel seed evaluation, and efficient GPU utilization. In practical deployments, additional gains can be achieved through early stopping and by filtering out clearly benign models during the early generations of Stage 1. This ensures that computational resources are focused only on adapters that exhibit non-trivial LoRA–base discrepancies or on LoRAs that do not quickly exhibit the reported functionality, making the auditing workflow feasible even when a platform must process many uploaded LoRAs.
> > >
> > > **On the rationale and audit scope of the objectives**
> > >
> > > **(1) On LoRAs that are highly semantic-aligned with corresponding trigger.**
> > > We agree that some LoRAs are semantically aligned with their trigger words. However, our goal is not to recover the actual trigger, but to detect the LoRA model-level behavioral shift introduced by the LoRA beyond what the base model can produce. Even for semantically aligned triggers, the LoRA typically encodes complex visual or stylistic details that cannot be replicated by simply appending a word to the prompt. As shown in our results, semantically related prompts do not reveal these behaviors, whereas the recovered activation key consistently exposes the LoRA-specific signal. This is exactly the forensic distinction our method is designed to capture.
> > >
> > > **(2) Small or non-semantic LoRA effects.**
> > > If a LoRA does not induce a meaningful semantic change for example, if it only adjusts pose, fixes hand artifacts, or applies slight aesthetic refinements then such an adapter is not harmful in practice, and it is not necessary for an auditor to recover a specific hidden concept. Our method intentionally focuses on coherent LoRA-induced signals that persist across seeds. When the encoded concept is indeed meaningful (identity, style, visual attribute), our objective reliably isolates it. Conversely, if the LoRA only introduces tiny or non-semantic shifts, the absence of a strong recovered key is itself an informative outcome, indicating that the LoRA does not encode harmful behavior.
> > >
> > > **(3) Style LoRAs.**
> > > For style-based LoRAs, the encoded modification is often mild, aesthetic, or harmless. If the stylistic shift is too small to produce a stable semantic signal, our method may not recover a distinct activation key. This is acceptable under our threat model, as such LoRAs do not pose the same risks as identity or concept-level LoRAs. When the stylistic change is substantial and coherent (e.g., a distinctive or heavily textured art style), the method succeeds in identifying it; when the change is minor, the lack of recovery simply reflects the fact that the LoRA itself does not introduce a strong semantic deviation.
> > >
> > > **Relation to multi-LoRA and composition in practice.**
> > > The “multi-task LoRA” setting discussed in the cited work [1] refers specifically to Trojaning text LoRAs during training, where multiple instruction-following behaviors are jointly embedded into a single adapter. In contrast, image LoRAs for diffusion models are typically trained to encode a single visual concept (identity, style, or an object), and “multi-LoRA usage” in practice almost always means combining multiple LoRAs at inference time, not multi-objective finetuning. As a result, the multi-task evasion concerns described for LLMs do not translate directly to image LoRAs. Our method is designed around the behavioral properties of image LoRAs in diffusion models, where concept shifts manifest directly in the generated images rather than in instruction-following behaviors.
> > >
> > > Nevertheless, following the reviewer’s instructions, we attempted to construct a multi-concept image LoRA and ran our full pipeline on it. The training set used for this experiment is shown in [dataset](https://postimg.cc/ygn1Hk2g), where we combined two heterogeneous concepts: a pixel-art style and a synthetic identity from [“this person does not exist.”](https://this-person-does-not-exist.com/en/download-page?image=gen38f3f5d02f18841ac9353f8db9445797) After training, the LoRA exhibited the expected issues, as when prompted with the real triggers it produced these [outputs](https://postimg.cc/G9RFc253), where the pixel-art concept is not learned as well in comparison to the identity. Importantly, when applying our activation-key discovery method, the recovered concept from the multi-concept LoRA is shown in [ouputs](https://postimg.cc/0rKvZkCG). The activation key successfully isolated the LoRA-specific concept despite the issues introduced during training, further supporting our argument that behavioral signals in generated images remain attributable even under mixed-concept finetuning.

---

### Official Review · Reviewer_qVw3 · 2025-10-26

**Soundness:** 3
**Presentation:** 2
**Contribution:** 3
**Rating:** 4
**Confidence:** 4

**Summary:**

This paper addresses LoRA adapter auditing by discovering "activation keys", a concept defined for text embeddings that expose hidden concepts in fine-tuned diffusion models. A two-stage framework is proposed, combining evolutionary search over discrete tokens (stage 1) with gradient-based refinement in continuous embedding space (stage 2). The objective function balances intra-model dispersion (LoRA outputs should be consistent) against inter-model similarity (LoRA should differ from base model). Experiments on six publicly available LoRAs are solid, demonstrating successful concept recovery, though with notable computational costs and ~22% failure rate.
This work aligns well with ICLR's core themes. Model auditing and interpretability are increasingly critical as generative models proliferate across the fields. The paper combines optimization theory, computer vision, and ML security in ways that should interest the ICLR community. The focus on LoRA adapters is particularly timely given their widespread deployment as a powerful PEFT method.

**Strengths:**

1. Important Problem: Auditing undocumented LoRA adapters matters. The community shares countless fine-tuned models, and yet many are without proper documentation. Having systematic ways to discover what they encode is genuinely useful.
2. Clean Problem Formulation: The activation key concept is well-defined in a straightforward manner. The objective function combining intra-model dispersion and inter-model similarity is intuitive and principled.
3. Practical Two-Stage Design: Starting with evolutionary search for coarse exploration, then refining with gradients makes sense. Stage 1 works in black-box settings; Stage 2 works with white-box access when available. This flexibility could be valuable.
4. Solid Empirical Validation: Testing on both stylistic and identity-based LoRAs shows generality. The quantitative metrics (CMMD, CLIP similarity) combined with qualitative VLM analysis provide multiple perspectives.

**Weaknesses:**

1. Objective Function Lacks Theoretical Grounding: Why should maximizing intra-LoRA dispersion while minimizing inter-model similarity necessarily recover the true concept? The paper doesn't provide a thorough theoretical justification. What guarantees exist that this objective aligns with finding semantically meaningful triggers rather than adversarial perturbations?
2. Limited Baseline Comparisons: The random prompt baseline is not a very strong case to be based upon. Why not compare against existing LoRA auditing methods mentioned in related work (Yao 2024's weight leakage, membership inference approaches)? Without stronger baselines, it's hard to assess whether the complexity of the two-stage framework is justified.
3. Computational Cost Is Prohibitive: Each experiment requires generating hundreds to thousands of images. On an A100, Stage 1 needs roughly 10×n×G images, Stage 2 adds about 1500 more. For practical auditing at scale, this cost seems problematic. The authors mention parallelization could help, but don't provide concrete timing comparisons or discuss computational efficiency as a design consideration.
4. Failure Rate Concerns: Combined ~22% failure rate across stages is perhaps not an ignorable number. The explanation attributes this to "initial random seeds in Stage 1," but this seems fixable. Why not run multiple Stage 1 initializations in parallel and select the best? The dependence on good initialization suggests the objective landscape could be better understood/designed.
5. Limited Discussion of False Positives: Can this method be fooled? What if someone deliberately creates a LoRA that appears benign under this auditing approach? The adversarial robustness of the framework isn't explored.
6. Evaluation Metrics Could Be Stronger: CMMD and CLIP similarity are reasonable but indirect. For identity-based LoRAs, why not consider adding face verification type of scores? For style LoRAs, perceptual metrics like LPIPS might be more appropriate. The VLM captioning is interesting, but perhaps can only serve as a qualitative (subjective) assistant judge.

**Questions:**

1. Can you provide a theoretical analysis showing your objective function provably recovers ground-truth concepts under reasonable assumptions?
2. What happens if you run multiple Stage 1 initializations? Does this reduce the failure rate proportionally, or are some LoRAs fundamentally harder to audit?
3. Have you considered testing adversarial scenarios where someone actively tries to evade your auditing method?
4. The objective function feels somewhat ad hoc. The authors don't justify the specific formulation of these terms beyond intuition. Why is Euclidean distance in CLIP space the right metric? Have you tried other divergence measures? Furthermore, the weighting parameters α and β are mentioned, but their values aren't specified. How sensitive are results to these choices? Ablation studies would strengthen this section. In addition, can you characterize when your objective succeeds? Even a toy model showing why dispersion+dissimilarity recovers concepts would help.

**Details Of Ethics Concerns:**

The authors do not explicitly discuss the ethical risks of the proposed method. Can an attacker utilize the method for an advanced attack or steal the functionality of the existing LoRA model?

---

> ### Author Response · Authors · 2025-11-20
>
> We sincerely thank the reviewer for the careful evaluation and for highlighting both the strengths and weaknesses of our work. We are grateful for the recognition of the problem importance, clean formulation, and practical design of the two-stage discovery framework. We address each concern in detail below.
>
> **On the theoretical grounding of the objective.**
> Our objective function follows directly from the key behavioral insight in activation-key recovery: when the same embedding is used to query both the LoRA-augmented model and the base model under identical seeds, any consistent asymmetry in their outputs must arise from the LoRA itself (as mentioned in Section 4). While this formulation is grounded in behavioral observations and practical intuition, providing formal theoretical guarantees remains challenging due to the non-convex, high-dimensional nature of diffusion models. Establishing theoretical links between this objective landscape and semantic ground-truth concepts is an interesting direction for future work. For now, our formulation offers an empirically validated and intuitively motivated approach for uncovering LoRA-induced concepts.
>
> **On computational cost.**
> The dominant runtime cost comes from evaluating multiple seeds per embedding, a process that is parallelizable. Additional engineering enhancements (e.g., distributed evolutionary search, caching, reduced diffusion schedules) could further reduce the cost if needed. Importantly, activation-key discovery is a forensic auditing task, where accuracy matters more than raw speed, and investigators typically analyze only a small number of suspect LoRAs. For these reasons, we believe the current runtime is practical, and any remaining overhead reflects implementation details rather than a fundamental limitation of the method.
>
> **On failure rate and initialization sensitivity.**
> The failure cases observed in our experiments stem from sub-optimal initial seeds during Stage 1. Because Stage 1 performs stochastic evolutionary exploration over a large token space, poor initialization can steer the population toward low-scoring regions. As the reviewer notes, simply restarting Stage 1 or running multiple initialization in parallel mitigates this issue, and our internal experiments confirm this. Once Stage 1 yields a reasonable embedding, Stage 2 consistently improves the objective score and alignment. We also note that our current implementation draws its token pool from the Brown corpus; using a more targeted or higher-quality corpus would likely improve initialization quality and further reduce failure cases. Exploring more effective or principled initialization strategies is an interesting direction for future work.
>
> **On adversarial evasion and false positives.**
> Our method can handle non-semantic triggers, as demonstrated by the synthetic-identity LoRA (LoRA 6, Fig. 3) with the trigger *“qwerty”*. To further validate this, we additionally trained a LoRA with a fully random trigger string, *rNbFUa06LF*, using a synthetic identity from *This Person Does Not Exist* (**[ground truth](https://this-person-does-not-exist.com/en/download-page?image=gen38f3f5d02f18841ac9353f8db9445797)**). The recovered activation key reliably elicited this hidden identity (**[output](https://postimg.cc/R3Nbd80y)**), showing robustness even under deliberately obfuscated triggers. In this work, our goal is to first demonstrate that activation-key forensic analysis is feasible under standard conditions. Adversarially crafted LoRAs designed to evade auditing are an important direction for future work. Such evasion strategies would effectively act as defenses against our method, but in practice would often reduce generative quality or hinder concept learning in turn raising the attacker’s cost and lowering the practicality of successful evasion. The need for deliberate obfuscation would itself indicate the strength and relevance of our approach.
>
> **On distance metrics and sensitivity to $\alpha,\beta,\gamma$.**
> Euclidean distance in CLIP space was used because CLIP embeddings correlate well with perceptual similarity and remain stable under gradient-based optimization. To study the contribution of each term in our objective, we included an ablation over $(\alpha,\beta,\gamma)$ in the revised submission (Section 6.2.2). This analysis had been omitted previously because we believed the importance of each term was evident from the formulation and from the intuition of capturing asymmetric behavior, but we now see the value in presenting it explicitly. The results show only the full objective recovered coherent activation keys across LoRAs and each individual term is insufficient on its own, confirming that compactness, divergence, and base-model spread jointly contribute to reliable discovery.
>
> We hope that our clarifications adequately address the concerns raised, and we are happy to provide any additional details if needed.

---

### Official Review · Reviewer_i8Ai · 2025-10-30

**Soundness:** 2
**Presentation:** 1
**Contribution:** 2
**Rating:** 2
**Confidence:** 3

**Summary:**

This paper focuses on the auditing challenges of undocumented LoRA adapters in text-to-image diffusion models, which can be used to inject sensitive or harmful concepts without disclosure. The authors introduce Activation Key, which is generated through a two-stage search framework, to reveal the concept only embedded in the LoRA adapter. Experiments on six public LoRA adapters show that the method effectively recovers the hidden concepts. Besides, the authors visualize the optimization trajectory with t-SNE, demonstrating how their method progressively creates a clear separation between the LoRA and base diffusion model.

**Strengths:**

+ Important research motivation.
+ Reasonable design of the auditing method.

**Weaknesses:**

- Poor writing and unclear methodology introduction.
- White-box method design is inconsistent with contributions.
- Insufficient experiment settings and evaluation.

**Questions:**

1. According to “Related Work”, one of your contributions is that the method is applicable in both white-box and black-box settings, while the stage-2 needs white-box access as demonstrated in subsec 6.2.2.

2. Many details of your method are difficult to understand and lack thorough introductions. What is the target to maintain the diversity within the base model in the training objective? How to achieve the token-level score $s_t$ of the prompt text based on $f(p)$, that is generated by image embedding?

3. The writing employs some non-standard terms and contains several incomplete sentences. For example, you use “spread” to denote $S_M$ without a detailed explanation; The second sentence in subsect 5.2 ends suddenly and unnaturally.

4. The experimental design lacks persuasiveness. You implement experiments only in six LoRA adapters without the large-scale datasets evaluation. Besides, the baselines are too limited and the effectiveness of “random prompt” is too weak as the baseline.

---

> ### Author Response · Authors · 2025-11-20
>
> We thank the reviewer for the constructive feedback and for highlighting both the motivation and the overall design of our auditing framework. We address each concern in turn.
>
> **On the applicability of the method in white-box and black-box settings.**
> Our framework is designed to operate in both access settings. Stage 1 functions purely in a black-box setting, using only model outputs to guide the search and requiring no access to parameters, gradients, or internal signals. This ensures that the method remains applicable whenever the investigator has only API-level interaction with the model. When white-box access is available, Stage 2 can further refine the embedding via gradients, which leads to improved performance. Such white-box access is common in many practical deployments for instance, LoRA platforms (e.g., CivitAI, HuggingFace Spaces) typically make model checkpoints available, and forensic or enterprise environments often operate with full model visibility. In this way, Stage 1 offers broad applicability under black-box constraints, while Stage 2 enhances performance in realistic settings where white-box access is available. We have added an ablation study in the updated version reporting the stand-alone performance of Stage 1 and Stage 2 across all six LoRAs (Appendix A.3). Together with the results in Fig. 8 and Fig. 9, these experiments show that Stage 1 alone already discovers meaningful activation patterns, while Stage 2 substantially improves performance when white-box access is available.
>
> **On the need for base-model diversity and clarification of the token-level score.**
> We use a base-model diversity term to ensure that the discovered activation key does not carry any semantic meaning in the base model and is unique to the LoRA model. This is crucial for auditing because an activation key that is meaningful to the base model would fail to isolate the LoRA-specific behavior we aim to reveal. By preserving intra-base diversity, we ensure that the optimized embedding activates only the LoRA-specific concept while remaining inert for the base model, consistent with our definition of an activation key. We have also updated the submission to include an ablation study for the hyperparameters in our objective, which highlights the importance of each term (Section 6.2.2). This analysis was omitted previously because we believed it was clear from the way the objective is defined and from the intuition behind the need to capture asymmetric behaviors. Regarding the token-level score, our calculation is done in the following way: for each prompt embedding, we compute the overall objective value, divide it by the number of tokens, and assign this normalized value uniformly to all tokens in the prompt. This serves as a heuristic ranking signal during Stage 1.
>
> **On non-standard terminology and writing clarity.**
> The term ``spread'' is commonly used and refers to the variance of the distribution, which is a standard statistical measure that we use to describe embedding dispersion, as explained in Section 4.1. We appreciate the reviewer identifying the incomplete sentence in Section 5.2; this was an editing oversight and has since been corrected in the updated version. We are happy to clarify any additional terms that may be unclear.
>
> **On experimental scale and baseline design.**
> We would like to highlight that the primary goal of this work is to investigate whether the activation-key discovery problem is even feasible under realistic forensic constraints, especially given that it is both a new and inherently difficult problem in which the investigator has no prior knowledge of the trigger or the training data. Because this problem is newly introduced, no large-scale or standardized dataset of LoRAs for auditing currently exists. Our six LoRAs span identity, style, and synthetic concepts, providing diverse behavioral patterns, and we view them as an initial benchmark for this emerging task. Despite the modest scale, these experiments demonstrate that the proposed method generalizes across structurally different LoRAs and that the underlying forensic mechanism is indeed achievable. With respect to baselines, existing approaches such as membership inference or training-data reconstruction are not applicable in our threat model, as the investigator has neither the trigger nor any knowledge of the LoRAs training data as mentioned before. Human probing is the closest comparable setting, which essentially corresponds to unguided random prompt exploration. Our ``random prompt’’ baseline therefore samples a fresh random prompt at each iteration for the same number of steps as our method, making it a reasonable and fair baseline until more specialized alternatives become available.
>
> We sincerely thank the reviewer for the helpful comments and the opportunity to clarify the methodology. We would be happy to provide any further details or answer additional questions.

---

> > ### Comment · Reviewer_i8Ai · 2025-11-25
> >
> > Thank you for your detailed replies and answering my questions.
> >
> > **Baseline：**
> > If the goal is to extract concepts that are unique in LoRAs, incorporating mixed or confusing training samples is necessary, rather than relying solely on clean and perfectly matched text–image pairs. This ensures that the concepts revealed by your method are genuinely meaningful and indeed correspond to the intended target (e.g., a specific political figure). Therefore, your approach falls within the scope of data auditing in LoRA if my understanding is right. The overall experimental setting appears similar to data auditing and backdoor trigger reconstruction. Consequently, including baselines from data auditing is appropriate, as the assumption of having an auditing target is reasonable in your work scenario.
> >
> > **Experiment:**
> > Understandably, introducing a new problem may lack an established and specialized dataset. However, experiment results with statistical significance are essential and feasible. For instance, train some  LoRAs and report the success rate with a quantified metric. This is more objective and convincing, rather than relying on a few examples. Without such systematic evaluation, it is difficult to rule out the possibility that the presented examples are hand-picked, favorable cases.

---

> > > ### Author Response · Authors · 2025-12-03
> > >
> > > We sincerely thank the reviewer for the follow–up comments. We address each point in turn.
> > >
> > > **Relation to data auditing and choice of baselines.**
> > > We agree that our setting is related to data auditing in the sense that the goal is to reveal concepts that originate from the LoRA rather than the base model. However, our problem formulation differs fundamentally from existing auditing and backdoor reconstruction techniques, as in our scenario, the investigator does not know what concept the LoRA encodes, has no access to any candidate identity images, and may not even know whether the LoRA contains a sensitive or hidden concept at all. Classical auditing methods assume that the auditing target is known in advance and evaluate membership or trigger success with respect to that known target. In our threat model, such baselines cannot be applied directly because there is no predefined identity or trigger on which to run these tests. For this reason, we focus on activation–key discovery itself as the core forensic mechanism as it produces an attributable concept purely from model behavior without requiring any prior knowledge of what the LoRA is intended to encode.
> > >
> > > **On “clean” vs. mixed training data and semantic meaningfulness.**
> > > We appreciate the concern about mixed or noisy training data. We note that all public LoRAs audited in this work are community trained, and we have no control over the cleanliness or curation of their datasets; any mixing, confounding, or label noise present in those LoRAs is therefore already reflected in our experiments. To additionally simulate an attacker who deliberately obfuscates the trigger or uses confusing training pairs, we trained two synthetic–identity LoRAs with non-semantic triggers, one using “qwerty” (Lora 6) and one using a fully random string as the trigger ***rNbFUa06LF***, using a synthetic identity sampled from *“This Person Does Not Exist.”*
> > > The ground-truth identity image is available
> > > **[here](https://this-person-does-not-exist.com/en/download-page?image=gen38f3f5d02f18841ac9353f8db9445797)**.
> > > The recovered activation key reliably elicited this hidden identity, as shown in **[Random LoRA Output](https://postimg.cc/R3Nbd80y)**. In both cases, our method successfully recovered the underlying concept, demonstrating robustness to intentionally obfuscated or non-semantic triggers. We also emphasize that an attacker who injects excessive noise into LoRA training will typically degrade the LoRA’s generative quality, making the model unstable and easier to flag. In such situations, the difficulty shifts to the attacker rather than the auditor. Our method therefore focuses on detecting coherent LoRA–specific concepts that remain stable across random seeds, regardless of how clean or mixed the underlying (unobserved) training data may be.
> > >
> > > **On statistical significance.**
> > > Our goal in this work is to demonstrate that activation–key discovery is feasible across diverse LoRA types. Accordingly, we evaluated identity, concept, and style LoRAs and showed that the method consistently uncovers the LoRA–specific concept in these different settings. We agree that a larger-scale evaluation would strengthen robustness claims and view it as a natural next step. However, we also emphasize that because the method relies on semantic signals extracted from generated images, it behaves consistently across LoRAs that encode concepts within the same broad semantic family. This is particularly evident for human faces, where individual identities whether public or privately trained still occupy a shared semantic space. In our additional tests on private identity LoRAs, the method recovered the underlying concepts in the same manner. This observation, together with the reported results, provides strong evidence that the technique generalizes well within a semantic domain. For this reason, while an exhaustive large-scale benchmark would undoubtedly be beneficial, the current experiments already demonstrate that the method is effective and stable across the diverse LoRA settings most relevant to the problem.

---

### Official Review · Reviewer_h4uh · 2025-10-31

**Soundness:** 2
**Presentation:** 3
**Contribution:** 2
**Rating:** 4
**Confidence:** 3

**Summary:**

This paper introduces the critical and underexplored problem of auditing undocumented Low-Rank Adaptation (LoRA) models. The authors formalize this as the "LoRA activation key discovery" task, where the goal is to find a text embedding that reliably triggers a LoRA's specific, fine-tuned behavior while remaining inert for the base model. To solve this, they propose a two-stage optimization framework: a black-box evolutionary search to find a promising initial prompt, followed by a white-box, gradient-based refinement of its embedding. The search is guided by a novel objective function designed to maximize the behavioral divergence between the LoRA and its base model. Experiments on six publicly available LoRA models show that the method can successfully recover the intended concepts.

**Strengths:**

1. Important Problem Formulation: The paper identifies and formalizes a critical, real-world problem concerning the safety, accountability, and auditing of community-shared generative models. This is a significant contribution to the responsible AI ecosystem.

2. Elegant Objective Function: The objective function, based on maximizing intra-LoRA consistency while minimizing inter-model similarity, is a principled and intelligent way to define the desired characteristics of an "activation key."

3. Principled Hybrid Search: The two-stage framework, combining evolutionary search for broad exploration and gradient-based methods for fine-tuning, is a strong and logical approach to the complex, hybrid search space.

**Weaknesses:**

1. Crucial Mismatch in Experimental Validation: The most significant weakness is that the experiments do not validate the method's utility for its stated purpose. The paper claims to uncover concepts "in the dark," but it is only tested on LoRAs with publicly known, non-adversarial triggers. This fails to demonstrate that the method can handle intentionally obfuscated, non-semantic, or compositional triggers that would be used in malicious LoRAs.

2. Insufficient Experimental Scale: The evaluation is conducted on only six LoRA models. While diverse in type, this small sample size is insufficient to make strong claims about the method's generalizability across the vast and heterogeneous landscape of community-trained LoRAs.

3. Practicality and Scalability Concerns: The proposed method, particularly the evolutionary search stage, is computationally expensive, requiring thousands of model inferences to audit a single LoRA. This raises serious questions about its feasibility for deployment at the scale required by model-sharing platforms.

**Questions:**

1. The core claim of the paper is to uncover hidden concepts, but the experiments were performed on LoRAs with public, semantically meaningful triggers. Could the authors provide evidence of their method's performance on a LoRA trained with a deliberately obfuscated or non-semantic trigger (e.g., a random string) to better support the central thesis?

2. Given the high computational cost, how do the authors envision this framework being practically deployed for large-scale auditing of thousands of models? Are there opportunities to significantly reduce the cost of the Stage 1 search?

3. The optimal performance relies on Stage 2, which requires white-box access. In a more realistic black-box (API-only) auditing scenario, what is the performance of the Stage 1 evolutionary search alone, and is it sufficient for reliable concept discovery?

---

> ### Author Response · Authors · 2025-11-20
>
> We sincerely thank the reviewer for the thoughtful and constructive feedback. We greatly appreciate the recognition of the problem formulation, the objective design, and the principled hybrid optimization framework. Below we address each concern in detail.
>
> **On evaluation with publicly known triggers.**
> We emphasize first that our evaluation already includes a non-semantic case, LoRA 6 (Figure 3) which is a synthetic-identity LoRA with a completely random string (*“qwerty”*) as the trigger, and our method successfully recovered the embedded concept despite the lack of semantic structure (Figure 4). To further address the reviewer’s suggestion, we additionally trained a LoRA with a fully random trigger, ***rNbFUa06LF***, using a synthetic identity sampled from *“This Person Does Not Exist.”*
> The ground-truth identity image is available
> **[here](https://this-person-does-not-exist.com/en/download-page?image=gen38f3f5d02f18841ac9353f8db9445797)**.
> The recovered activation key reliably elicited this hidden identity, as shown in **[Random LoRA Output](https://postimg.cc/R3Nbd80y)**. This demonstrates that our method remains effective even when the trigger is intentionally obfuscated and entirely non-semantic.
>
> **On experimental scale.**
> We would first like to clarify that the central contribution of this work is to establish whether activation-key forensic analysis is even feasible in the first place. This is both a new and inherently difficult problem, as the investigator must operate under a zero-knowledge setting where the trigger words, training data, and intended concept of the LoRA are all unknown. Because the problem itself has not been previously formulated, the field currently lacks prior work, established methodologies, or standardized datasets for evaluating activation-key discovery. Our study therefore focuses on six diverse LoRAs covering identity, style, and synthetic concepts, including the non-semantic Synthetic Identity LoRA, to investigate whether hidden fine-tuned concepts can be systematically recovered at all. Despite the modest scale, the consistent recovery across such structurally different LoRAs provides strong initial evidence that this forensic task is solvable. As a natural next step, we plan to curate a larger collection of LoRAs to build the first broader benchmark for future large-scale evaluation.
>
> **On practicality, runtime, and scalability.**
> We would like to point out that the runtime of our method can be substantially reduced through standard parallelization techniques. In particular, the main cost comes from evaluating multiple seeds per embedding, which is fully parallelizable. With GPU batching, all seeds are processed in one pass, reducing generation to ∼3 seconds per image. Further optimizations (e.g., caching CLIP embeddings, fewer diffusion steps, parallel Stage 1 workers) can reduce runtime even more. At the same time, we believe that in forensic auditing settings, accuracy and reliability matter far more than speed, and investigators typically will examine only a small number of suspect LoRAs. The current runtime is therefore practical, and any remaining overhead is an implementation detail rather than a fundamental limitation of the method.
>
> **On Stage 1 performance in black-box settings.**
> It is important to note that our method is designed to operate effectively under black-box conditions. Stage 1 relies solely on input-output behavior and does not require access to model parameters, intermediate activations, or gradients. Our experiments confirm that Stage 1 is capable of discovering meaningful and usable activation patterns even without any white-box information in most cases. We have added an ablation study in the updated version that reports the stand-alone performance of Stage 1 and Stage 2 across all six LoRAs (Appendix A.3). As shown in Fig.8 and Fig.9, Stage 1 already induces a noticeable divergence between LoRA and base-model behaviors, demonstrating that it remains practically useful for strictly black-box, API-only auditing. At the same time, when white-box access is available, which is a realistic assumption in many operational settings such as LoRA platforms (e.g., CivitAI, HuggingFace Spaces) and enterprise deployments where full checkpoints are typically accessible the Stage 2 provides an additional improvement in performance. Our method therefore reaches its full potential when both stages are available. In this sense, our framework is robust in black-box scenarios while naturally yielding stronger results whenever full model visibility is available.
>
> We sincerely thank the reviewer for their thoughtful feedback and constructive insights. We appreciate the positive assessment of our problem formulation and methodology, and we hope that our clarifications address the raised concerns. We would be happy to provide any further details or answer additional questions.

---

### Meta-Review · Area_Chair_sMpT · 2025-12-11

**Summary:**

Reviewers found the problem important and the two-stage framework promising, but they raised concerns about limited evaluation, noting the small set of six LoRAs, lack of statistical analysis, and weak baselines that make robustness and generality hard to assess. Several questioned the theoretical grounding of the three-term objective, asking for clearer justification of why it should recover true concepts—especially for stylistic or low-impact LoRAs—and requested ablations and sensitivity studies.
Another major concern was scalability and practicality, given the method’s high computational cost and notable failure rate, which reviewers felt undermines its feasibility. Concerns were raised regarding the strength of the “black-box” claim since the best results rely on white-box Stage 2 and reviewers also noted missing comparisons to related methods and noted significant presentation issues (unclear explanations, terminology).

**Reviewer Concerns:**

Some concerns were partially addressed in the rebuttal. The authors added experiments on non-semantic and fully random triggers, which helps support claims about recovering “hidden” concepts. They also clarified the intended distinction between black-box Stage 1 and white-box Stage 2, added ablations quantifying their contributions, and explained details of the objective. Additional discussion of computational parallelization and practical deployment addressed some questions about runtime, and the authors provided more intuition for the three-term objective and its applicability across LoRA types.

However, several issues remain even after the rebuttal. The core limitations regarding evaluation scale, statistical rigor, and weak baselines were not meaningfully resolved. Theoretical concerns also remain as reviewers questioned whether the objective is principled or guaranteed to recover semantic concepts, and the rebuttal provided intuition but no deeper justification. Practical feasibility is still uncertain. While engineering optimizations were mentioned, no concrete runtime benchmarks or reductions in the reported failure rate were demonstrated. Importantly, some reviewers explicitly stated post-rebuttal that the responses did not fully address their core concerns, especially regarding baseline adequacy and practical deployment.

Overall, while the rebuttal clarified several methodological points and added helpful ablations, the key concerns emain only partially addressed.

**Reviewer Scores:**

Reviewer h4uh (score 4) raised concerns about limited experimental scale, lack of truly hidden triggers, and questions about scalability and black-box practicality. The rebuttal addressed several points with new experiments on non-semantic/random triggers and added ablations clarifying Stage 1 performance and feasibility assumptions. However, the core issues of small-scale evaluation and unresolved practical deployment challenges remain. The reviewer’s score is unlikely to change substantially.

Reviewer i8Ai (score 2) was concerned about unclear methodological explanations, inconsistent claims about white-box vs. black-box applicability, weak baselines, limited experimental design, and writing issues. The rebuttal clarified many technical points, added ablations, and corrected presentation issues, but the reviewer explicitly stated post-rebuttal that fundamental concerns—especially weak baselines, lack of statistical evaluation, and insufficient empirical rigor—remain unaddressed. The reviewer appears unconvinced and unlikely to raise the score.

Reviewer qVw3 (score 4) highlighted limited theoretical justification, weak baselines, computational cost, failure rate, and insufficient discussion of adversarial robustness. The rebuttal provided conceptual clarifications, ablations, and additional tests on obfuscated triggers, partially addressing these concerns. Still, the main issues—lack of theoretical grounding, high computational cost, and weak baselines—remain open, making a score increase unlikely.

Reviewer BCTv (score 4) raised concerns about scalability, the validity and scope of the three-term objective, lack of ablations, unclear black-box applicability, and vulnerability to multi-task or evasive LoRAs. The rebuttal added ablations, clarified scope, and included a mixed-concept LoRA experiment. However, the reviewer’s follow-up comment made clear that concerns about scalability and objective validity remain unresolved. A score increase appears unlikely.

Reviewer ffWZ (score 6) was generally positive but noted limited experimental breadth, weak baselines, and missing ablations. The rebuttal addressed the methodological concerns and added ablations, but the experimental-scale issues were not resolved. Still, given the already positive stance, the reviewer would likely maintain but not raise the score.

---

### Decision · Program_Chairs · 2026-01-26

Reject